# The Application of Biomass-Based Catalytic Materials in the Synthesis of Cyclic Carbonates from CO_2_ and Epoxides

**DOI:** 10.3390/molecules25163627

**Published:** 2020-08-10

**Authors:** Li Guo, Ran Zhang, Yuge Xiong, Dandan Chang, Haoran Zhao, Wenbo Zhang, Wei Zheng, Jialing Chen, Xiaoqin Wu

**Affiliations:** 1Key Laboratory of Hubei Province for Coal Conversion and New Carbon Materials, School of Chemistry and Chemical Engineering, Wuhan University of Science and Technology, Wuhan 430081, China; xyg201903@163.com (Y.X.); cdd110925899@163.com (D.C.); Zhaohr290370@163.com (H.Z.); wustzhangwenbo@163.com (W.Z.); zhengwei321@126.com (W.Z.); 2Hubei Key Laboratory of Biomass Fibers and Eco-dyeing & Finishing, Wuhan Textile University, Wuhan 430073, China; rzhang@wtu.edu.cn

**Keywords:** CO_2_, cyclic carbonate, biomass, heterogeneous catalysis, coupling reaction

## Abstract

The synthesis of cyclic carbonates from carbon dioxide (CO_2_) and epoxides is a 100% atom economical reaction and an attractive pathway for CO_2_ utilisation. Because CO_2_ is a thermodynamically stable molecule, the use of catalysts is mandatory in reducing the activation energy of the CO_2_ conversion. Considering environmental compatibility and the high-efficiency catalytic conversion of CO_2_, there is the strong need to develop green catalysts. Biomass-based catalysts, a type of renewable resource, have attracted considerable attention due to their unique properties—non-toxic, low-cost, pollution-free, etc. In this review, recent advances in the development of biomass-based catalysts for the synthesis of cyclic carbonates by CO_2_ and epoxides coupling are summarized and discussed in detail. The effect of biomass-based catalysts, functional groups, reaction conditions, and co-catalysts on the catalytic efficiency and selectivity of synthesizing cyclic carbonates process is discussed. We intend to provide a comprehensive understanding of recent experimental and theoretical progress of CO_2_ and epoxides coupling reaction and pave the way for both CO_2_ conversion and biomass unitization.

## 1. Introduction

The burning of carbonaceous fuel and intense human activities have caused a sharp increase in CO_2_ emissions more than 35 gigatonnes per year [1]. Huge CO_2_ emissions may lead to global climate problems and rising sea levels, which can seriously threaten the human living environment. CO_2_ is also deemed an abundant and cheap C1 source to produce chemicals and energy [2]. However, due to the most oxidized carbon atom being in CO_2_ and its chemical inertness, the conversion of CO_2_ usually requires harsh reaction conditions, such as high temperature, high-pressure of CO_2_, and the need for catalysts [3,4,5,6].

In the past decades, massive efforts have been focused on catalyst design and optimization of reaction conditions for CO_2_ conversion. Cyclic carbonates are widely used as important chemical intermediates for pharmaceutical production, plastics, resin materials, and outstanding solvents for battery electrolyte, among others [6]. Although the mechanism of the reaction between CO_2_ and activated epoxides relies upon the intramolecular nucleophilic substitution to the saturated carbon as the key step (see Scheme 1), the overall process can also be regarded as a formal [3 + 2] cycloaddition. Thus, the resulting cyclic carbonates are obtained with a 100% atom economy. Several representative catalysts, including ionic liquids [7,8,9,10], metal complexes [11,12,13], metal oxides [14,15], porous carbon [16,17,18], porous organic polymers [19,20,21,22], metal-organic frameworks (MOFs) [23,24,25], covalent organic frameworks (COFs) [26,27,28], and biomass-based catalysts [29,30,31], have been developed for the synthesis of cyclic carbonates from CO_2_ and epoxides. In particular, biomasses including cellulose, lignin, and hemicellulose, among others, are the most accessible and renewable resources with abundant hydrogen bonds, which can be used as catalyst materials for CO_2_ conversion. By designing and functionalizing biomass-based materials, the biomass-based catalyst can be fine-tuned, making it a promising material for CO_2_ and epoxides coupling reactions [32].

According to the literature, the synthesis of cyclic carbonates from CO_2_ and epoxide is usually composed of several steps: (1) the activation of epoxides and ring-opening; (2) the CO_2_ inserts into the oxygen anion intermediate; (3) the ring-closing reaction inside the molecule (Scheme 1) [33,34,35,36,37]. Fortunately, biomass-based materials serve as promising catalysts because of their unique advantages: (1) Biomass materials are usually rich in hydroxyl groups; previous researches have shown that hydrogen-bond donors (HBD) such as cellulose, chitosan, phenol-based compounds, amino alcohol, and amino acid can promote the CO_2_ and epoxides coupling reaction by activation of the epoxide [6]. (2) Biomass materials also contain abundant functional groups; for example, lignin, the second most naturally plentiful biopolymer substance, is rich in functional groups such as phenol hydroxyl, alcoholic hydroxyl, methoxy group, aldehyde group, ketone group, and carboxyl group [38]. By designing and functionalizing the biomass materials, the physical and chemical properties of biomass can be fine-tuned, making them promising materials for CO_2_ capture and conversion. (3) The cheap, renewable, non-toxic, and good biocompatibility of biomass materials can greatly facilitate their industrial application in the synthesis of cyclic carbonates from CO_2_ and epoxides. All of these advantages make biomass-based catalysts ideally suited for the coupling reaction of CO_2_ and epoxides.

The employment of biomass-based catalysts in CO_2_ has undergone three development stages. To begin with, the biomass/co-catalyst systems were explored. As the biomass exhibits many hydroxyl groups that can be used as the hydrogen-bond donor, the synergic catalytic effect between the biomass and the co-catalyst (i.e., KBr, KI, TBAI, TBAB, and ZnBr_2_) was carefully studied [39]. The second stage focused on the development of functionalized biomass-based catalysts. It is important to recognize that the biomass materials have a limited amount of reactive sites, most of them Lewis acid centers. However, the coupling reaction of CO_2_ and epoxides needs not only Lewis acid centers to activate the epoxides, but also Lewis base centers to activate CO_2_ to accelerate the CO_2_ conversion. To solve this problem, some basic groups (i.e., ionic liquid, organic amine, and Schiff-base) have been introduced into the biomass materials [40]. In the third and most recent stage, attempts have been made to explore and expand the porous carbons using biomass as a carbon source [41]. Thus, herein, we present a summary and discussion of the development of biomass-based catalysts for direct synthesis of cyclic carbonates from epoxides and CO_2_.

## 2. The Biomass-Based Catalyst Systems for CO_2_ Conversion

### 2.1. Biomass/Co-Catalysts Systems

Although many non-biomass based catalysts could serve as dominating systems for cyclic carbonate synthesis, expensive catalysts and transition metal wastes are the main drawbacks from a green chemistry viewpoint. Biomass with many kinds of HBD and in cooperation with various nucleophiles as binary catalysts are a greener alternative. In early researches, HBD organocatalytic systems were used in the selective coupling of CO_2_ and epoxides into cyclic carbonates. Table 1 lists and compares catalytic systems and reaction conditions reported in the literature. As mentioned above, the HBD group (e.g., -OH, -COOH, -NH-) can bind to the oxygen atom of epoxide by forming hydrogen bonds, resulting in polarization of the C-O-ring bond, which could facilitate a nucleophilic attack by the halide and the ring-opening (Scheme 1). For instance, Hou et al. reported that using the pentaerythritol/KI catalyst system, -OH in the pentaerythritol could interact with O in epoxide, therefore facilitating its ring-opening by nucleophilic attack of a halide anion (I^−^) [42]. Meanwhile, pentaerythritol and KI could form a complex and then expose the iodide ion. Suitable nucleophilicity and leaving ability of the halogen anion played an important role in the CO_2_ and epoxides coupling reaction. Mirza et al. also screened a series of co-catalysts with pentaerythritol and found that the pentaerythritol/nBu_4_NI system had a 96.0% yield (Table 1, entry 2) under mild reaction conditions [43], which were comparable with the transition-metal based catalyst. Besides, Zhang et al. found that phenolic compounds showed excellent performance on the CO_2_ and epoxides coupling reactions [44]. The results revealed that the activity and selectivity of 1,2-benzenediol were better than that of phenol. The main reason for this phenomenon may be the steric hindrance effect. Density functional theory (DFT) studies provided additional support for the hydrogen bond-promoted fixation of carbon dioxide and epoxides in cyclic carbonates. The reaction mechanism of the non-catalyst process and nBu_4_NBr/1,2-benzenediol-catalyzed process and the relative energy of the rate-determining step were carefully investigated (Figure 1). As shown in Figure 1, it demonstrated that the fixation of CO_2_ with ethylene oxide proceeded via hydrogen bond with a much lower barrier for the rate-determining step, when compared to the non-catalytic process or the nBu_4_NBr-catalyzed process. The notable activity for the nBu_4_NBr/1,2-benzenediol dual catalytic system possibly originated from the cooperative actions of 1,2-benzenediol and nBu_4_NBr, which helped to activate the epoxides and more easily stabilize the intermediates and transition states through hydrogen bond interactions, making the reaction much easier. Similar results were obtained by Kleij et al. with pyrogallol as the catalyst (Table 1, entry 5) [45]. It was found that 1,2-benzenediol and pyrogallol showed a higher yield than phenol, which meant that the polyhydroxy system could play vital roles in stabilizing the intermediates and transition states. Moreover, -COOH and -NH- groups were also used in the CO_2_ and epoxides coupling reaction, which displayed a considerable catalytic activity (Table 1, entries 6–9) [46,47,48]. Interestingly, amino acids, the natural, green, and non-toxic chemicals have shown a higher TON (turnover number) and TOF (turnover frequency) in the coupling reactions of CO_2_ with various epoxides (Table 1, entry 7). The higher catalytic activity may be ascribed to the -NH- group. The -NH- group in the amino acids not only activated epoxide by hydrogen bonding but also activated CO_2_ by -NH_2_ [47]. Although these HBD organocatalytic systems can accelerate the CO_2_ and epoxides coupling reaction, it is difficult to separate the substrates from the catalyst.

As discussed above, the synergistic effect between hydroxyl groups and co-catalysts plays an important role in the synthesis of cyclic carbonates from CO_2_ and epoxides. Based on this conclusion, abundant, renewable, environmentally benign, and biocompatible biomasses, as a natural biopolymer in all kinds of hydroxyl groups, can act as an alternative HBD in this reaction. In 2010, the cellulose/KI catalytic system was first used in the coupling reactions of CO_2_ and propylene oxide [29]. It is worth pointing out that the cellulose/KI catalytic system was also very efficient for other epoxides, producing corresponding cyclic carbonates with yields of 92–99%. The higher yield of cyclic carbonates in the cellulose/KI catalytic system may be due to the existence of numerous hydroxyl groups on the vicinal carbons of cellulose, which could be the main reason for the high efficiency of cellulose to accelerate the reactions (Scheme 2). There are four types of species formed by diols and PO: (**a**) one of the two OH groups forms a hydrogen bond with PO and the other is free; (**b**) each of the OH group forms a hydrogen bond with a PO; (**c**) the free hydrogen of the OH group forms a hydrogen bond with PO, while the other forms an intramolecular hydrogen bond to form a ring species; (**d**) two OH groups form seven-membered ring species (Scheme 3). These catalytic activity dates suggested that the H became more active after the intramolecular hydrogen bond was formed, leading to high efficiency for accelerating the CO_2_ and epoxides coupling reaction.

As in previous studies, the two hydroxyl groups in 1,2-diols can form a ring species with other atoms by hydrogen bonding and the 6-membered or 7-membered ring species are most stable [49,50]. As a result, the adjacent hydroxyl groups in the cellulose can form seven-membered ring species with PO by hydrogen bonding (Scheme 3d), which results in higher efficiency of the cellulose. After Han’s work, Zhang et al. compared various superbase/cellulose catalytic systems (including 1,8-diazabicyclo[5.4.0]-undec-7-ene (DBU), 7-methyl-1,5,7-triazabicyclo[4.4.0]dec-5-ene (MTBD), 1,5,7-tri-azabicyclo[4.4.0]dec-5-ene (TBD), 1,4-diazabicyclo[2.2.2]octane (DABCO), *N*-methylimidazole (MIm), imidazole (Im), *N*,*N*-dimethylaminopyridine (DMAP), pyridine (Py), monoethanolamine (MEOA), diethanolamine (DEOA), triethanolamine (TEOA), triethylamine (TEA), and diethylamine (DEA)) in the chemical fixation of CO_2_ into cyclic carbonates [51]. It was found that DBU was the most excellent organic base in the conversion of PO in the presence of cellulose under metal-free and halide-free conditions. Based on Heldebrant et al.’s work, the pK_a_ value of the organic bases decreases in the order of TBD > MTBD > DBU>TEA > DMAP > Py> DEA > MEOA > DEOA, DABCO > TEOA > MIm, Im [52]. However, the activity order of the above bases is not in strict accordance with the established pK_a_ that is DBU > DMAP > DABCO, TEOA > DEOA, MIm >MEOA > TEA > DEA, Py > MTBD > Im, TBD. Hence, both the basicity of the base and the steric hindrance as well as synergistic effect of the hydroxyl group in cellulose can influence catalytic activities (Scheme 4). Park and co-workers further realized that carboxymethyl cellulose (CMC), a derivative of the cellulose family with the carboxymethyl group (-CH_2_COOH) bound to some of the hydroxyl groups of the glucopyranose monomers, would be able to promote the CO_2_ and epoxides coupling reaction [53]. A variety of ionic liquids immobilized on the carboxymethyl cellulose (CMIL) was applied in the synthesis of PC from CO_2_ and PO, and the 1-butyl-3-triethoxysilylpropyl imidazolium iodide (IL-4-I) with CMC showed the highest catalytic activity and selectivity (Table 2, entry 3). The excellent activity of these catalysts may be derived from the potential synergistic capability of -COOH and -OH groups on the CMC along with the X^−^ ions in the medium to eventuate the coupling reaction of PO with CO_2_. According to the DFT analysis, a typical portion of the CMC structure comprising three anhydroglucose units (AGU) with one -CH_2_COOH group present was used as shown in the inset of Scheme 5. When the ring opens to form the ring-opened product complex, the negative charge is on the O-atom (carbonyl) of the carbonate and competes for the H-atom of the carboxyl group. The H-atom stays closer to the carbonyl oxygen with a bond distance of only 1.43 Å, which is lesser than usual hydrogen bond distances, indicating a stronger interaction with the product intermediate (Scheme 5). The -COOH groups on the support material are shown to stabilize the cyclic carbonate formed, thereby promoting the reaction. The probable mechanism is that the -OH groups in CMC act as Lewis acidic sites to coordinate with the O atom of epoxides, which can activate the epoxy rings. Next, the nucleophilic Cl^−^ attacks the less-hindered C atom of epoxy rings to open the ring. Then, CO_2_ molecules insert the opened epoxy ring to interact with the oxygen anion and obtain an alkylcarbonate salt, which can be stabilized by the -COOH groups in CMC. At the same time, the ring closes to produce cyclic carbonates and CMC is recovered. Moreover, Hou and co-workers disclosed CMC supported imidazolium-based ionic liquids (ILs), coupling with a series of Lewis acid catalysts system for the synthesis of cyclic carbonates [54]. As is known, the supported catalysts more or less face the problem of catalyst loss during recyclability of the catalysts. However, the CMC supported with both hydroxyl groups functionalized IL (1-hydroxypropyl-3-*n*-butylimidazolium chloride, HBimCl) and NbCl_5_ exhibited high catalytic activity and showed excellent leach-resistant property.

Besides, chitosan (CS), the most abundant natural biopolymer, also has excellent properties, such as biocompatibility, biodegradability, non-toxicity, and good adsorption properties [56]. Compared to cellulose, CS contains one amino group and two hydroxyl groups in the repeating hexosaminide residue, which could be considered as not only a much more suitable alternative amino-functionalized polymer for active CO_2_ under suitable conditions, but also a much more suitable hydrogen bonding donor for active epoxide rings. Zhang et al. reported that CS/1-ethyl-3-methyl imidazolium bromine (EMImBr) facilitated the reaction better than cellulose/EMImBr, which can be attributed to the activation of CO_2_ by the tertiary amine group in CS [30].

As mentioned above, pyrogallol/nBu_4_NI systems display a significant synergistic catalytic effect in the CO_2_ and epoxides coupling reaction [43]. Lignin, the second most naturally abundant biopolymer substance in plant cell walls, is exceeded only by cellulose. Lignin is mainly composed of *p*-hydroxyphenyl (H), guaiacyl (G), and syringyl (S) (Scheme 6), which has many phenolic structures in the skeleton [31]. Liu and co-workers first used lignin as a catalyst in the chemical fixation of carbon dioxide with epoxides to produce cyclic carbonates [55]. It was demonstrated that the enzymatic hydrolytic lignin (EHL) and KI catalyst system showed a 93% yield of PC with no significant drop in the yield of PC after five successive cycles. Furthermore, Guo and co-workers proposed a strategy to facilitate the construction of a green, stable, and efficient catalyst based on the extraction of lignin [31]. Soda lignin with high purity (99.3%) and abundant hydroxyl groups (6.49 mmol·g^−1^) was obtained by a membrane separation technique and used in the synthesis of cyclic carbonates from CO_2_ and epoxides. The synergistic effect between aliphatic-OH, phenolic-OH, and carboxylic-OH in SL may play an important role in the CO_2_ and epoxides coupling reaction. The dimeric intermolecular hydrogen bonding interaction may also account for the higher catalytic activity in the formation of cyclic carbonates under mild conditions (Scheme 7).

From the view of green chemistry, constructing biomass-based catalyst systems can not only convert CO_2_ into value-added chemicals but also provide an ideal green process for the utilization of “carbon-neutral” resources.

### 2.2. Functionalized Biomass-Based Catalysts

Although biomass-based catalysts can greatly enhance the activity for cyclic carbonate synthesis, functional groups that can activate CO_2_ and facilitate the ring-opening steps are still lacking. Therefore, some attention has been focused on functionalized biomass-based catalysts. Park et al. first reported the quaternization of ethylenediamine-functionalized celluloses (en-Cell) by the microwave irradiation method [57]. Moreover, microwave quaternized celluloses (mQCs) were also used in the synthesis of cyclic carbonates under solvent-free conditions (Scheme 8). It was found that the yield of PC was 97% under a low catalytic loading (0.4 mol% mQCs). The mQC-1.I showed a TON of 242, which was superior to most earlier reports on biopolymer-based heterogeneous catalysts (Table 3, entry 2). Based on the structural analysis of the mQC-1.I, the excellent catalytic activity could be attributed to the increased accessibility of the twin hydroxyl groups toward epoxide co-ordination and the greater CO_2_ diffusivity. The use of microwaves furnished a higher BET surface area of 50–71 m^2^g^−1^, which is 20–30 times higher than that of the starting material, fibrous microcrystalline cellulose. Hence, the mQC-1.I exhibited excellent catalytic performance in the transformation of CO_2_ and epoxides to form cyclic carbonates, and they could be reused more than six times without any significant loss of activity.

Many kinds of ILs and IL-modified heterogeneous catalysts have been able to achieve the goal of chemical fixation of CO_2_ with excellent product yield and selectivity under mild reaction conditions [65]. Einloft and co-workers developed cellulose-based poly(ionic liquids) (CPILs) and used them as heterogeneous catalysts for CO_2_ chemical transformation into cyclic carbonates (Scheme 9) [58]. It was worth mentioning that the use of ZnBr_2_ as a co-catalyst was able to increase PC yield in a CPIL-TBP catalytic system. This can be attributed to the high reactivity of Zn with bromide nucleophilicity facilitating the epoxide ring-opening. Additionally, Jia et al. designed a novel metal-free cellulose-based Schiff-base heterogeneous catalyst (Cell-H_2_L) for the ring-opening addition reaction of CO_2_ with epoxides to synthesize cyclic carbonates for the first time (Scheme 10) [39]. When using the Cell-H_2_L/TBAB catalytic system, PO was quantitatively converted to PC in 6 h with a catalyst load of only 0.4 mol% (Yield = 99%, TON = 505, T = 100 °C, p(CO_2_) = 30 bar). The excellent catalytic activity might be derived from the synergistic effects of OH groups and Lewis basic sites. A DFT study on the Schiff-base ligand in the CO_2_/PO system showed that the phenolic -OH groups in the Schiff-base complex could deliver a hydrogen bond with PO at a length of 1.787 Å (Scheme 11). After the hydrogen-bond formation between PO and phenolic -OH, the length of the C-O bonds in epoxide increased from 1.435 to 1.451 Å and 1.434 to 1.445 Å, respectively. Moreover, the DFT simulation results showed that after interacting with H_2_L, the linear CO_2_ bond angle changed from 180° to 177°, which resulted in the polarization, and the CO_2_ got activated.

In recent years, bio-renewable CS has also been exploited, due to its physicochemical characteristics and bioactivities and the property of being easily modified chemically or physically. He and co-workers first reported a functionalized biopolymer-chitosan-supported quaternary ammonium salt (CS-N^+^R_3_X^−^, Scheme 12) [59]. This single-component active catalyst was applied in a variety of terminal epoxides, producing corresponding cyclic carbonates at yield of 98% and selectivity of 99%. It was the first time to realize the synthesis of carbonates from CO_2_ and epoxides without any additives or co-catalysts by an active and readily recyclable single-component catalyst. Thereafter, Zhang et al. developed chitosan functionalized 1-ethyl-3-methyl imidazolium halides (CS-EMImX, X = Cl, Br) for this reaction, and found that the catalyst could exhibit a high PC yield (96%), which was comparable to that of the homogeneous EMImBr (PC yield 96%) [30]. The hydrogen bond-assisted ring-opening of epoxide and the nucleophilic tertiary nitrogen-induced activation of CO_2_ in the CS-EMImBr catalytic system may play bi-functional roles in promoting the reaction. Similar results were obtained by Park et al. with quaternized chitosan (QCHT) as a catalyst for the coupling reaction of CO_2_ and epoxides [60]. It was found that QCHT afforded 86% yield at 120 °C and 11.7 bar CO_2_ in 6 h (Table 3, entry 8). The main advantage of the QCHT catalyst was the great potential for industrial applications due to the advantages of stability, low cost, and simple separation from the product. Besides, Park et al. also developed a microwave irradiation method to prepare the quaternization of chitosan (QCHT) [61]. The effects of different alkyl chain length and anion of the QCHT catalyst on the coupling reactivity were carefully studied. The obtained results suggest that the positive inductive (+I) effect increased, leading to a decrease of the effective positive character of the quaternized species; consequently, the reactivity of QCHT catalysts decreased when moving from methyl to butyl. Moreover, the high nucleophilicity and leaving ability of these anions resulted in higher catalytic activity. As a result, the quaternized chitosan with iodomethane (QMMI) exhibited the highest activity, with an 89% yield of PC (Table 3, entry 9). Previous research had displayed that the carboxyl-functionalized phosphonium-based ionic liquids performed well in CO_2_ chemical transformation into cyclic carbonates with 97.3% (TOF = 65 h^−1^) yield of PC [66]. For this reason, Luo and co-workers further prepared chitosan-grafted quarternary phosphonium ionic liquid (1-butyl-triphenylphosphonium bromide, denoted hereinafter as CS-[BuPh_3_P]Br) by a simple method. This CS-[BuPh_3_P]Br showed high activity, indicating the synergistic effect of OH, P cations, and Br^−^ (Scheme 13) [62]. To start with, the C-O bond was polarized by hydrogen bonding, which facilitated the opening of the epoxide ring. In addition, polarization (due to hydrogen bonding), electronic interaction (due to phosphonium cation), and a nucleophilic attack by a bromide anion took place. Finally, the haloalkoxy intermediate further reacted with CO_2_ to form a linear halocarbonate that transformed into cyclic carbonate through intramolecular substitution of the bromide anion. Furthermore, the chitosan-based catalytic system containing both carboxylic acid and quaternary ammonium moieties (abbreviated as CBCN) was developed by Caillol et al. (Scheme 14) [63]. PC was obtained with very high yield in a few hours under mild conditions (7 bar, 80 °C) and without a solvent. Based on these early researches, it is noted that if the interaction between K^+^ and I^−^ is low, the anion is much freer to react with the electrophilic species. In this system, the modification of chitosan can improve the dissociation of potassium iodide, which allows the synthesis of PC under low pressure, low temperature, with a low amount of potassium iodide, and without solvent. 

Recently, porous metal-organic frameworks (MOFs) showed high capacity for CO_2_ capture, which was associated with their large surface areas and pore volumes [67]. However, their uncertain behavior at high temperature hampers their industrial use. For this reason, Sobral and co-workers synthesized a chitosan-based meso-tetrakis(4-sulfonaophenyl)porphyrin (CS-TPPS) for adsorption and catalytic conversion of CO_2_ [64]. CS-TPPS displayed the adsorption capacity of 0.9 mmol CO_2_/g compared to the adsorption capacity of 0.05 mmol CO_2_/g of pure chitosan and adsorption capacity of 0.2 mmol CO_2_/g of pure TPPS (Scheme 15). The adsorption capacity demonstrated the interactions between CO_2_ and the heteroatoms (N, O, S), and an increase in selectivity for the adsorption of CO_2_ in porous polymers. In consequence, the PC yield of CS-TPPS was 66%, as compared to 31% with pure chitosan as a catalyst at 75 °C for a reaction time of 6 h. Similar results were obtained by Ghiaci et al. with the composite of chitosan and a dicationic ionic liquid (Scheme 16) [40]. It was found that the Ch-ILBr had a mesoporous structure with a surface area of 57 m^2^g^−1^, pore volume of 0.16 cm^3^g^−1^, and average pore diameter of 2.48 nm. They suggested that chitosan through hydrogen bonding, coordination of amine group with CO_2_, and also loosely bonded bromide ion to imidazolium ion, had a synergistic effect on high yield and selectivity of cyclic carbonates under mild conditions. Based on previous studies, it was found that metal-free catalysts were generally considered less active than transition metal-based catalysts, which encouraged researchers to develop new biomass-based composite catalysts. Then, Boroujeni et al. designed and synthesized metallophthalocyanines on chitosan (MPcs@CS; M = Cu, Ni, Co) in deep eutectic solvents (Scheme 17) [32]. It was shown that CuPcs@CS with TBAB can significantly improve catalytic activity for the coupling of styrene oxide (SO) with CO_2_ under mild conditions (80 °C, 1 bar). These results indicate a visibly synergistic effect between the chitosan and metallophthalocyanine. In general, all of the studies may open new vistas towards developing new biomass-based efficient materials for a wide range of environmental applications, particularly in CO_2_ capture and conversion.

### 2.3. Biomass-Derived Porous Carbons Materials

Carbon materials have been demonstrated to be one of the most promising adsorbents for CO_2_ capture due to their outstanding features, such as easy accessibility from natural raw materials, excellent thermal and chemical stability, high surface area, high amenability for pore structure modification and surface functionalization, and ease of preparation and regeneration [68]. Among precursors used for the synthesis of porous carbons, biomass feedstocks are competitive because they are not only abundant, sustainably renewable, and cost-effective, but also exhibit extremely high CO_2_-sorption capacities. For example, coconut shell-based porous carbons were found to adsorb almost 5 mmol g^−1^ of CO_2_ at 25 °C and over 7 mmol g^−1^ of CO_2_ at 0 °C, which can be ascribed to its high microporosity and nitrogen content [69]. However, the uptake capacity of CO_2_ is relatively low because it is a weak physisorption process. Therefore, many efforts have been focused on incorporating basic nitrogen groups into the carbon framework to enhance the adsorption of CO_2_, which can significantly improve the surface polarity, electrical conductivity, and electron-donor capacity of the carbon materials. For example, Hu et al. developed *N*-doped porous monolith with outstanding CO_2_ capture ability and conversion performance (Scheme 18) [70]. Alginic acid (AA), a commercially available biomass material, was subjected to hydrothermal treating in the presence of pyrrole, ethylenediamine, and glutaraldehyde at 180 °C and followed by freeze-drying and annealed processing. AA-950, which meant that the pyrolysis temperature was 950 °C, showed high catalytic activity (84.3% yield of chloropropene carbonate and TOF: 1768). These experimental results suggest that the presence of Lewis base sites in its framework and its high CO_2_ adsorption could enhance the synthesis of cyclic carbonates from CO_2_ and epoxide. Additionally, using pyridine as a reasonable model catalyst for DFT study suggested that the reaction occurred through a heterolysis of the epoxide C-O bond activated by the nucleophilic attacked of pyridine and formed a zwitterion (Int1, Scheme 19) with an anion located at the oxygen (O_e_) and a positive charge delocalized at the pyridine ring. Then, the O_e_ rapidly attacked the carbon of CO_2_ to generate another zwitterion (Int2) with a carbonate moiety. Subsequently, the cyclization proceeded via a five-membered cyclic transition state, leading to a cyclic carbonate (product) and a regenerated pyridine. A similar mechanism may happen with the AA-950, but the reaction barrier may be lowered by the stabilization of zwitterions due to the bigger delocalization of positive charge via a larger conjugated system.

According to pioneering work, a great number of synthesis strategies have been developed to prepare porous materials, including surfactant templating, colloidal crystal templating, polymer templating, and emulsion templating, among others [71]. Islam et al. prepared a nitrogen-doped mesoporous carbon material (*N*-GMC) by using glucose as a carbon source and melamine as a nitrogen source using the soft templating (Brij-35) method [72]. It was shown that a 98% yield of styrene carbonate could be obtained at 80 °C under 10 bar of CO_2_. The high yield of cyclic carbonates can be ascribed to the high surface area and basic nitrogen in the porous *N*-GMC. In a work on renewable *N*-doped active carbon, Konwar and co-workers prepared the *N*-doped porous carbon by one step phosphoric acid activation of nitrogen-containing biomass precursors (chitosan, *Jatropha curcas*, and *Mesua ferrea* DOWCs) [73]. These *N*-doped active carbon samples were referred to as CA500 (chitosan), JA500 (*Jatropha* DOWC based), and MA500 (*Mesua* DOWC based) respectively. The high cyclic carbonate yield of the optimum carbocatalysts (MA500 and CA500) may be attributed to the Lewis basic sites originating from pyridinic, pyridonic, and quaternary *N*-sites, which can activate the CO_2_ molecule (Scheme 20). As a result, the plausible mechanism can be closely approximated by a Lewis-base (pyridine, guanidine) catalyzed mechanism, with the basic pyridinic, pyridonic, and quaternary N acting as active sites for CO_2_ activation and cyclic carbonate formation.

To further reduce the production cost of porous carbon materials, a lot of interest has been paid to the transformation of bio-waste into carbon materials. For example, Kong et al. synthesized a class of nanoporous carbons by the direct carbonization of bio-waste cow manure, followed by activation with KOH and NaNH_2_ [74]. The synthesized cow manure-derived carbons (CMCs) had a BET surface area as high as 1106 m^2^g^−1^, hierarchical nanopores, and nitrogen sites with abundant pyridinic characteristics successfully doped into their networks and were seen to be highly efficient and selective in the capturing of CO_2_. The activated CMCs were then supported with Co^2+^ and used as heterogeneous catalysts for CO_2_ conversion. It was found that Co/CMC displayed excellent performance in the catalytic conversion of CO_2_, with a yield of >95%, selectivity of >99%, and TOFs of ~700 h^−1^ at 100 °C and 50 bar for 1.5 h (Table 4, entries 6 and 7). Recently, Jiang and co-workers developed biomass-derived hierarchically porous carbon materials by facile carbonization of the g-C_3_N_4_ and chitosan mixture in N_2_ [75]. The nitrogen-decorated hierarchically porous carbons (*N*-HPC-xs) had a unique helix 2D carbon network with abundant micro-meso-macropores structure, which showed excellent performance for CO_2_ capture (2.05–3.38 mmol g^−1^ at 0 °C, 1.0 bar, and 1.52–2.61 mmol g^−1^ at 25 °C, 1.0 bar) with high IAST (ideal adsorbed solution theory) CO_2_/N_2_ selectivity (83.8–111.3 at 0 °C and 71.2–88.2 at 25 °C). The DFT results suggested that the pyridinic nitrogen species had much higher binding energy (−24.14 kJ mol^−1^) than the pyrrolic (−16.9 kJ mol^−1^) and graphitic (−18.96 kJ mol^−1^) nitrogen species. However, the binding energy values of CH_4_ on pyridinic, pyrrolic, and graphitic nitrogen species were −8.19, −6.84, and −7.93 kJ mol^−1^, which meant that the adsorption capacity of CH_4_ or N_2_ was much weaker than CO_2_ in *N*-HPC-xs. Moreover, on functionalization with either Co or Zn, the resulting Co or Zn@*N*-HPC-xs showed high catalytic efficiency in CO_2_ cycloaddition to epoxides for the formation of carbonates (Table 4, entries 8 and 9). Table 4 summarizes biomass-derived porous carbons materials for the chemical fixation of CO_2_ into cyclic carbonates.

## 3. Summary and Outlook

This review highlights significant progress in biomass-based materials as highly selective and efficient catalysts for the coupling reaction of CO_2_ and epoxides. Using biomass-based catalysts can not only provide feasible pathways for improving the value of biomass, but also achieve the goal of CO_2_ capture and conversion. However, direct utilization of biomass as a catalyst usually needs high temperatures and pressure with low TON or TOF. In order to overcome the shortage of harsh reaction conditions and low TOF, some functionalized biomass-based catalysts and biomass-derived porous carbon catalysts have been designed and applied in the chemical fixation and conversion of CO_2_. Despite impressive progress in biomass-based catalysts, there are still issues that need to be resolved for sustainable development of cycloaddition catalysts. Most important, there are certain challenges in CO_2_ capture and conversion. A feasible method is to improve CO_2_ adsorption sites by introducing some Lewis basic sites and tuning the stability of the intermediates. Moreover, the BET surface area of biomass materials is usually low, which significantly limits their application as biomass-based materials. Although the carbonization processing of biomass may be an alternative way of improving the surface area, finding a suitable method to retain the HBD groups and increasing the BET surface of catalysts still need to be further investigated. Finally, the TON or TOF of the biomass-based catalysts are lower when compared to the MOF or COF materials. As a result, designing biomass-based catalysts in synergistic activation of CO_2_ and epoxides is another huge challenge. Without a doubt, there is a bright future for biomass-based catalysts as a green alternative in CO_2_ fixation and conversion, and this review will give some inspirations for the design and construction of biomass-based catalysts.

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
