# Peer review of "The Application of Biomass-Based Catalytic Materials in the Synthesis of Cyclic Carbonates from CO2 and Epoxides"

_molecules, 2020, doi:10.3390/molecules25163627_

Round 1

Reviewer 1 Report

the publication is a review on the use of CO2 in combined epoxy opening / protection transformations in the form of carbonates.
The authors provide an exhaustive literature on the subject with an interest in non-metallic catalysts. This is part of an approach to exhibit the most environmentally friendly conditions. Different catalysts obtained from biomass are exposed as well as the catalytic mechanisms.
The references cited are relevant and the literature cited is very recent.
There are some small errors in the text, the authors will be able to find them on the attached file.

some questions :
scheme 5: the activation of CO2 is not clear in scheme 5. Is there any H bond between CO2 and the polysaccharide? what is the nature of the bond between CO2 and the epoxide?

concerning the relation between pKa and the catalytic activity of different bases (page 5). The authors compare two sets of pKa taken in two different solvents. If they compared pK and catalytic activity of the two families of pk independently, a correlation can be established.

Author Response

Response to Reviewer 1 Comments

Point 1: The activation of CO2 is not clear in scheme 5. Is there any H bond between CO2 and the polysaccharide? What is the nature of the bond between CO2 and the epoxide?

Response 1: The referee raised a very good question. An explanation of scheme 5 has been amended in Line 188-196. The added description in the revised manuscript is as follows: “The -COOH groups on the support material are shown to stabilize the cyclic carbonate formed, thereby promoting the reaction. The probable mechanism is that the -OH groups in CMC act as Lewis acidic sites to coordinate with the O atom of epoxides, which can activate the epoxy rings. Next, the nucleophilic Cl- attacks the less hindered C atom of epoxy rings for opening the ring. Then, CO2 molecules insert the opened epoxy ring to interact with the oxygen anion and obtain an alkylcarbonate salt, which can be stabilized by the -COOH groups in CMC. At the same time, the ring is closing to produce cyclic carbonates and CMC is recovered”.

On the other hand, there is actually no hydrogen bond between CO2 and the polysaccharide in this catalytic system. If CO2 molecules insert the opened epoxy ring, the new C-O bond is formed between CO2 and the epoxide.

Point 2: Concerning the relation between pKa and the catalytic activity of different bases (page 5), the authors compare two sets of pKa taken in two different solvents. If they compared pK and catalytic activity of the two families of pK independently, a correlation can be established.

Response 2: Thanks a lot for such valuable suggestions. We have carefully studied the relationship between pKa and catalytic activity in the two sets of pKa taken in two different solvents. However, there is no obvious correlation between pKa and catalytic activity. The pKa values of the organic bases in acetonitrile decrease in the order of TBD>MTBD>DBU>TEA>DMAP>Py. However, the activity order of the above bases is not in strict accordance with the established pKa, which is DBU>DMAP>TEA>Py>MTBD> TBD. Similarly, the pKa values of the organic bases in water decrease in the order of DEA>MEOA>DEOA, DABCO>TEOA>MIm, Im. But the activity order of the above bases is also not in strict accordance with the established pKa, which is DABCO, TEOA>DEOA, MIm>MEOA>DEA>Im.

Reviewer 2 Report

This manuscript contains an interesting review of catalytic processes to obtain cyclic carbonates, produced by reaction of CO2 with different epoxides. The interest of this reaction is justified as a procedure to value CO2 as a chemical reagent, given the excess production that is currently generated by industry and transport. Perhaps this review should also indicate why the compounds obtained by the coupling reaction of CO2 with epoxides may be important. Can they suppose a massive consumption of CO2, or would it be fine chemical products with a high added value?

On the other hand, in the title of the manuscript, and throughout it, the process of synthesis of cyclic carbonates is described as a process of "cycloaddition". A cycloaddition mechanism, according to chemical terminology, implies that it is developed through a "concerted" process, which occurs in a single step, with the simultaneous movement of three pairs of electrons (if the reaction occurs under thermal control). But here a stepwise mechanism is proposed (Scheme 1), with a first step of activating the epoxide, in which the attack of the halide anion occurs, followed by a second step in which a nucleophilic attack is generated to the carbon atom CO2 and the halide removal. Therefore, the term cycloaddition should be removed from the title of the manuscript, replacing it with "synthesis of cyclic carbonates", or "CO2 and epoxides coupling reactions", or any other that the authors like more, but that makes it clear that it is not a cycloaddition process.

In this sense, this term should also be removed from the manuscript, where it appears collected more than thirty times. This, despite the fact that the term cycloaddition appears in the title of the references 15,18,26,30,31,34,35,37,40,43,54,61 and 62, since the inappropriate application of a chemical expression does not justify continuing to repeat it, especially when there are correct options to indicate the reaction described.

Author Response

Response to Reviewer 2 Comments

Point 1: Perhaps this review should also indicate why the compounds obtained by the coupling reaction of CO2 with epoxides may be important. Can they suppose a massive consumption of CO2, or would it be fine chemical products with a high added value?

Response 1: Thanks a lot for such valuable suggestions. We have added some descriptions (Line 40-44) in the section of introduction to explain the application of the products obtained by the coupling reaction of CO2 with epoxides. The added description in the revised manuscript is as follows: “Cyclic carbonates are widely used as important chemical intermediates for pharmaceutical production, plastics, resin materials, and outstanding solvents for battery electrolyte, and so on [6]. The CO2 and epoxides coupling reactions are the 100% atom economical reaction to obtain cyclic carbonates.”

Point 2: On the other hand, in the title of the manuscript, and throughout it, the process of synthesis of cyclic carbonates is described as a process of "cycloaddition". A cycloaddition mechanism, according to chemical terminology, implies that it is developed through a "concerted" process, which occurs in a single step, with the simultaneous movement of three pairs of electrons (if the reaction occurs under thermal control). But here a stepwise mechanism is proposed (Scheme 1), with a first step of activating the epoxide, in which the attack of the halide anion occurs, followed by a second step in which a nucleophilic attack is generated to the carbon atom CO2 and the halide removal. Therefore, the term cycloaddition should be removed from the title of the manuscript, replacing it with "synthesis of cyclic carbonates", or "CO2 and epoxides coupling reactions", or any other that the authors like more, but that makes it clear that it is not a cycloaddition process.

In this sense, this term should also be removed from the manuscript, where it appears collected more than thirty times. This, despite the fact that the term cycloaddition appears in the title of the references 15,18,26,30,31,34,35,37,40,43,54,61 and 62, since the inappropriate application of a chemical expression does not justify continuing to repeat it, especially when there are correct options to indicate the reaction described.

Response 2: I’m quite sorry for the inaccurate and grossly misleading in the depiction of "cycloaddition". The title has been changed from “The application of biomass-based catalytic materials in the CO2 cycloaddition reaction” to “The application of biomass-based catalytic materials in the synthesis of cyclic carbonates from CO2 and epoxides”. Some inaccuracies in the text have also been corrected (Line 67, 77, 79, 116, 120, and so on).
